biomaterials

round bamboo, polyethylene glycol treatment, paraffin heat treatment, cracking resistance, outdoor exposure

**Authors for correspondence:**
Hui Wang
e-mail: wh-snowis@outlook.com
Fangli Sun
e-mail: sun-fangli@163.com

# Combination of polyethylene glycol impregnation and paraffin heat treatment to protect round bamboo from cracking

Jin Rao[1], Jun Jiang[1], Nayebare Kakwara Prosper[1], Xiushu Yang[1], Tingsong Liu[1], Wei Cai[2], Hui Wang[1] and Fangli Sun[1]

[1]School of Engineering, Zhejiang A&F University, Lin'an 311300, People's Republic of China
[2]Anji Zhujing Bamboo Technology Co., Ltd, An'ji 313300, People's Republic of China

JR, 0000-0003-0680-9786; FS, 0000-0003-3487-7399

Round bamboo has drawn more and more attention in architecture, decoration and recreational products. Splitting brings some safety problems, which shorten the service life of round bamboo. In this paper, three schemes were adopted as follows to solve the problem: round bamboo was impregnated in polyethylene glycol (PEG)-1000 solution alone, heat treatment in paraffin alone or treated with the combination of PEG impregnation and paraffin heat treatment (PEG–PH). The treated bamboo was exposed outdoors for 26 weeks to evaluate the development of cracks. Results showed as follows: the initial split of PEG–PH-treated bamboo appeared after 22 weeks, while that of the controls after 2 weeks, the total length of cracks was 2271.31 and 873.5 mm for the control and PEG–PH-treated bamboo, respectively. To reveal the reasons for reduced cracks, scanning electron micrograph (SEM) was employed to observe the microstructure of bamboo; besides, hydrophobicity of bamboo was characterized by the water contact angle. PEG can swell the cell wall and the better hydrophobicity of round bamboo could be achieved after PEG–PH treatment. Therefore, the combination of PEG immersion and paraffin heating is an effective and practicable method in bamboo treatment, especially for round bamboo with high moisture content.

## 1. Introduction

As an abundant, environmentally friendly and renewable resource [1], bamboo has been widely used in various fields. Owing to its

hard, smooth and glossy surface, as well as the special cultural connotation, round bamboo has found more and more applications in architecture [2,3], such as tourist resorts, exhibition halls and leisure parks. However, splitting, decay and mildew are the main problems threatening round bamboo structures, and greatly shorten their service life. Decay and mildew arise from fungi, which can be restrained by commercial fungicides, while splitting is hard to prevent owing to the poor penetration of chemicals into the bamboo. Smoking treatment is an effective method for round bamboo protection, but it is time consuming as was reported at least three months of treatment. Surface coatings offer temporary protection for round bamboo and are widely used in constructions, but bring a bad impression on the application due to the short service life. The successful application of chemicals in wood stabilization has offered inspiration and experiences to bamboo crack control treatment. Acetic anhydride, furfuryl alcohol and hydroxyethyl methylacrylate were used to increase the dimensional stability of bamboo [4,5], as well as neem seed oil treatment [6]. Most of these modifications were conducted on the middle part of the culm wall, and at a lower moisture content to achieve efficient penetration. However, as for structural round bamboo with epidermis and inner part, 7–8 m long, and an average moisture content higher than 60%, the permeability of chemicals is crucial to the effect of modification. Owing to the high moisture content, the widely used vacuum pressure treatment for chemical penetration is inefficient for round bamboo.

Polyethylene glycol (PEG) as a bulking agent for the wood cell wall was frequently used to improve the dimensional stability of wood, especially in maintaining the ancient wood building with high moisture content [7–9]. Jeremic et al. [10] found that PEG could penetrate the wood cell wall and displace water in it. The presence of PEG keeps the wood cell wall in a swollen state and prevents shrinkage, accordingly, diminishing the risk of cracking [11,12]. It was reported that the low molecular weight of PEG easily penetrated wood cell walls and restrained the shrinking of the cell wall [13]. Stamm [14] insisted that PEG with molecular weights of 1000 (PEG-1000) was most suitable to prevent the surface checks of wood. Stamm [15] found that spruce (*Picea asperata* Mast.) saturated with 25% solutions of PEG-1000 gave an anti-shrink efficiency (ASE) of 100%. PEG is typically used to impregnate wood with high moisture content, permeating through concentration difference, which is probably suitable for round bamboo treatment, especially fresh ones with high moisture content.

Owing to the special structure [16], round bamboo is susceptible to cracking when using conventional high-temperature drying technology. Therefore, it is necessary to develop a drying technology which is suitable for round bamboo. Paraffin had been used for decades as a surface protector for wood because it retarded the permeation of water from the air by establishing a predominantly hydrophobic barrier at the wood/air interface [17,18]. In addition, paraffin with various boiling points and flash points can be used as a heating medium.

This paper combined the impregnation of PEG and paraffin heat treatment in which the bulking of round bamboo was achieved by the penetration of PEG through concentration difference, followed by the paraffin drying procedure, which at the same time, formed a paraffin coating. The dimensional stability of round bamboo was improved by the above methods.

# 2. Material and methods

## 2.1. Materials

Freshly cut 5-year-old *Phyllostachys iridescens* was obtained from Anji, Zhejiang, China. The round bamboo with a diameter of 42–61 mm was cut into specimens with a length of 300 mm. The average moisture content of the specimens was $58.3 \pm 5\%$, and six replicates were prepared for each treatment. To avoid large deviation in every group, the diameter of four samples was in the range of 42–50 mm with one node, and the other two were in the range of 50–61 mm with two nodes, one at each end.

PEG with an average molecular weight of 1000 and a melting point of around 40°C was purchased from Shanghai Sinopharm Chemical Reagent Co. Ltd, China. Semi-refined paraffin, with a melting point of around 58°C, was bought from PetroChina Co. Ltd.

## 2.2. PEG treatment (PEG)

The specimens were immersed in 25 wt% solution of PEG-1000 for 24 h at room temperature and atmospheric pressure. After that, the specimens were air-dried at room temperature for 24 h.

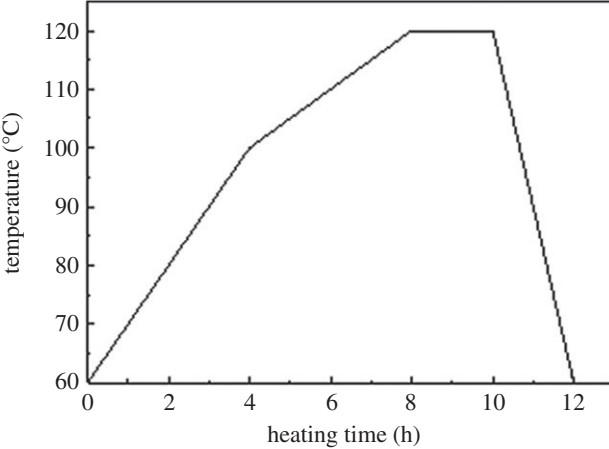

**Figure 1.** Heating schedule used for round bamboo.

## 2.3. PH treatment

The solid paraffin was first heated at 60°C until melted, then the specimens were immersed into the melted paraffin and heated according to figure 1. To avoid cracking caused by high heating rate, the temperature was increased from 60 to 100°C at a speed of 10°C h$^{-1}$, then a speed of 5°C h$^{-1}$ from 100 to 120°C. Round bamboo would develop splits during excessive high temperatures owing to its closed hollow structure. According to the preliminary study, the heating temperature adopted in this paper did not exceed 120°C.

## 2.4. The combination of PEG and PH treatment (PEG–PH)

Specimens were firstly treated with PEG as described in §2.2, then immersed in melted paraffin and heated according to the method in §2.3.

## 2.5. Retention of PEG

Owing to the high moisture content of fresh round bamboo, the bulking coefficient was difficult to determine according to the commonly adopted formula [7]. Acetone has a lower boiling point, dissolves PEG easily and is insoluble in paraffin; hence, it can be used to extract PEG from bamboo. Referring to the method from Kang *et al.* [19], specimens treated with PEG and PEG–PH were soaked in acetone for two weeks, followed by air-drying at room temperature for 24 h and then oven-dried for 24 h. The retention of PEG ($R$) was then calculated according to equation (2.1). Because round bamboo is not strictly regular in shape and thus difficult to get the accurate volume, so the retention of PEG was expressed as the weight ratio of PEG in round bamboo, which is different from the calculation of retention in ASTM D 1413–99 [20].

$$R\left(\text{g kg}^{-1}\right) = \frac{m_0 - m_1}{m_0} \times 100,$$  (2.1)

where $R$ is the retention of PEG, calculated as the weight ratio of PEG in bamboo (g kg$^{-1}$), $m_0$ is oven dry weight of specimen after PEG or the combination of PEG and paraffin treatment (g), $m_1$ is dry weight of acetone-extracted specimens (g).

## 2.6. Test on crack resistance

Specimens treated in different ways were placed on the open balcony of the National Center for Comprehensive Utilization of Wood Resources in Zhejiang A&F University without covering, as shown in figure 2. Bamboo strips with the thickness of 5 mm were put under the specimens to avoid getting in contact with water on heavy rainy days. The experiment lasted for 26 weeks from 1 June 2017 to 30 November 2017. The information about temperature, humidity, precipitation and sunshine during this period was plotted in figure 3.

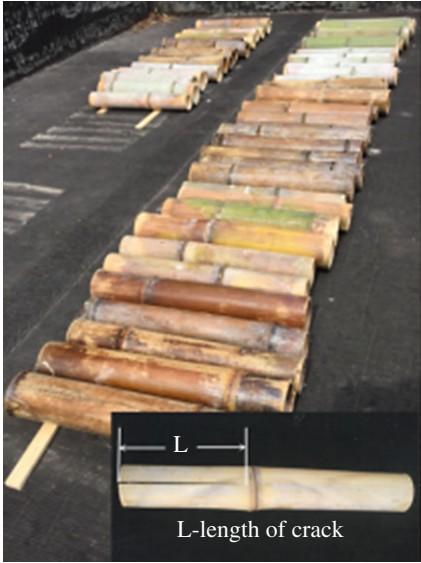

**Figure 2.** Outdoor experiment and method for measuring the length of the crack.

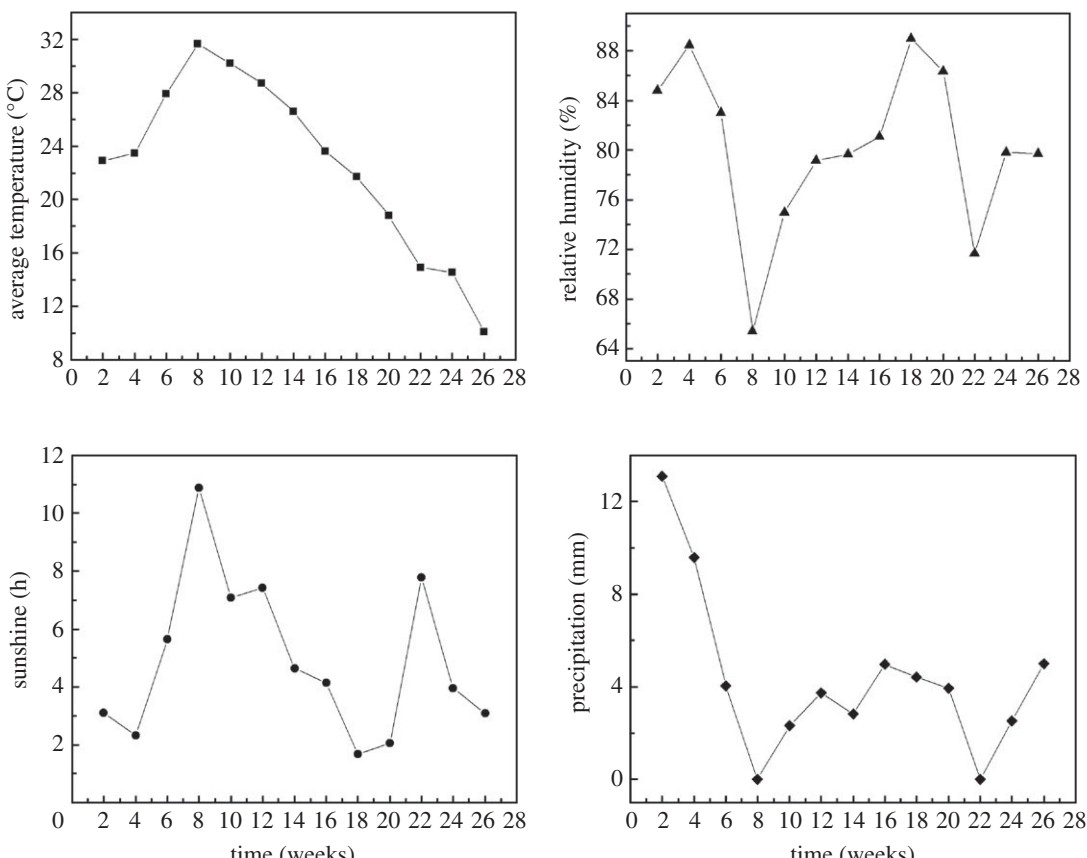

**Figure 3.** The change of temperature, humidity, precipitation and sunshine during testing.

There was no standard test method for evaluating the cracks or splits of round bamboo. What is more, the width and depth of cracks were difficult to measure during the outdoor exposure, because they varied with the fluctuation of humidity, and sometimes the cracks extended rapidly through the culm wall once developed. Therefore, the only index which could be used to evaluate the degree of cracking was the length of crack as determined according to figure 2.

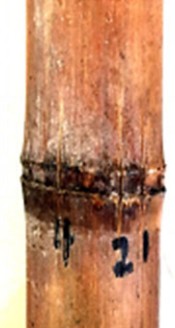

**Figure 4.** Three cracking rules of round bamboo.

## 2.7. Determination of surface contact angle

Hydrophobicity of the bamboo transverse section was characterized by water contact angle (WCA) measurements (Dataphysics OCA50, Stuttgart, Germany). The WCA values were acquired at room temperature with a water droplet volume of 5 µl placed on the surface of the specimens, and the average of five measurements taken at different positions on each specimen was applied to calculate the final WCA angle.

## 2.8. Morphology

To determine the distribution of PEG and paraffin in round bamboo, as well as the morphology of bamboo after 26-week outdoor exposure, samples were prepared by hand and microtome and imaged by SEM (Hitachi SU-8010, Tokyo, Japan) with an accelerating voltage of 15 kV.

# 3. Results and discussion

## 3.1. Type of crack in round bamboo

There are three major types of cracks in specimens, including end cracking, node cracking and thorough cracking as shown in figure 4. The frequent and serious cracks mainly originated from end checks, which were ascribed to the rapid loss of water from the end of the bamboo.

## 3.2. The content of PEG

Paraffin heat treatment can form a hydrophobic surface for PEG-treated round bamboo but probably cause the loss of PEG due to the movement of PEG in water evaporation. Therefore, the amount of PEG will be determined. The average content of PEG was 17.0 (±0.8) g kg$^{-1}$ for PEG treatment alone, while 15.3 (±1.7) g kg$^{-1}$ for the combination of PEG and paraffin treatment. It seemed that PEG was slightly lost after the paraffin heat treatment, which was probably caused by the movement of PEG with the evaporation of water.

## 3.3. Effect of different treatments on bamboo cracking

### 3.3.1. Cracks on the control

Cracks developed quickly in the control bamboo (figure 5). Eight cracks appeared after two weeks of outdoor exposure, and the largest one was as long as 132.4 mm. The number of cracks increased from 8 to 10 during four weeks experiment and continued to increase to 13 at the eighth week. No new cracks appeared during the next five weeks, then followed a rapid increase in the number from 16 to 19. The length of most cracks grew slowly and was less than 155.3 mm. After 18 weeks, a 278.6 mm crack appeared. At the end of the experiment, the total number of cracks was 19, with the lengths between 12.2 and 158.0 mm except two thorough cracks.

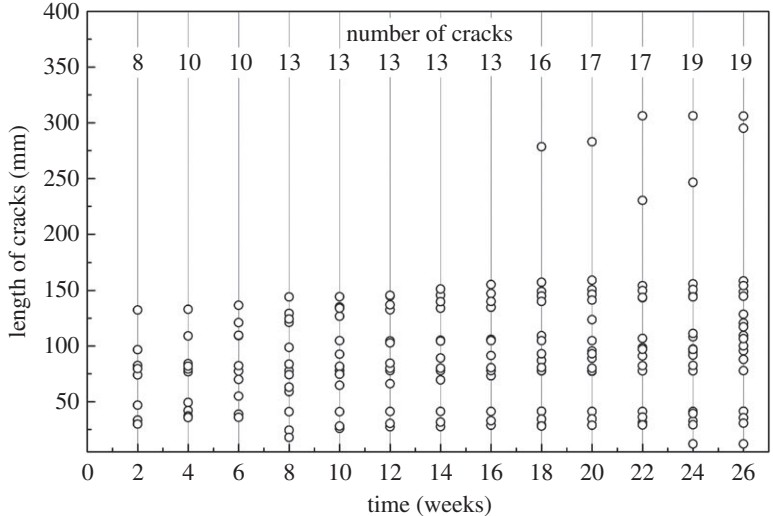

**Figure 5.** Development of cracks on the controls during 26-week outdoor exposure.

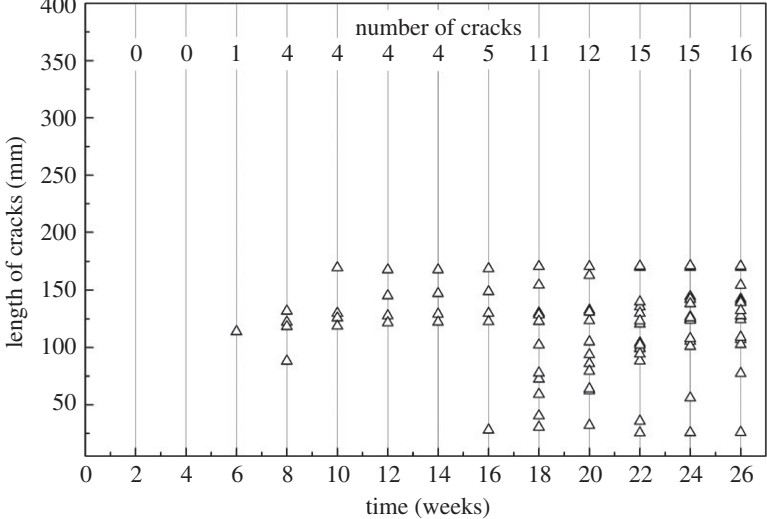

**Figure 6.** Development of cracks on the PEG-treated bamboo during 26-week outdoor exposure.

### 3.3.2. Cracks on PEG-treated round bamboo

PEG could diffuse into the fresh bamboo and replace water by concentration difference, which will provide dimensional stability to bamboo. No cracks were observed in the PEG-treated round bamboo during the first four weeks exposure as compared to the 10 cracks in the control (figures 5 and 6), which suggesting that PEG could restrain the presence of cracks by maintaining the dimension of culm wall. A crack appeared after six weeks experiment and followed by four cracks at the eighth week. After 16 weeks, the number of cracks developed rapidly, which might have been caused by the loss of PEG on rainy days, but the length of cracks kept below 128.27 mm during the whole test duration. The total number of cracks reached 16 at the end of the experiment, and the length of cracks was all less than 170.92 mm and no thorough cracks were noted. The above analyses demonstrated that PEG-treated round bamboo could restrain crack to a certain extent.

### 3.3.3. Cracks on paraffin-treated bamboo

Paraffin is a harmless mixture of higher alkanes with stable chemical properties and is widely applied in lumber and wood-based panels as waterproofing material [21]. It is reported that cracks always occur when oven-drying round bamboo at 120°C [19], but in our experiment, no crack was produced during the heat treatment of round bamboo in paraffin at the same temperature. Paraffin-treated bamboo

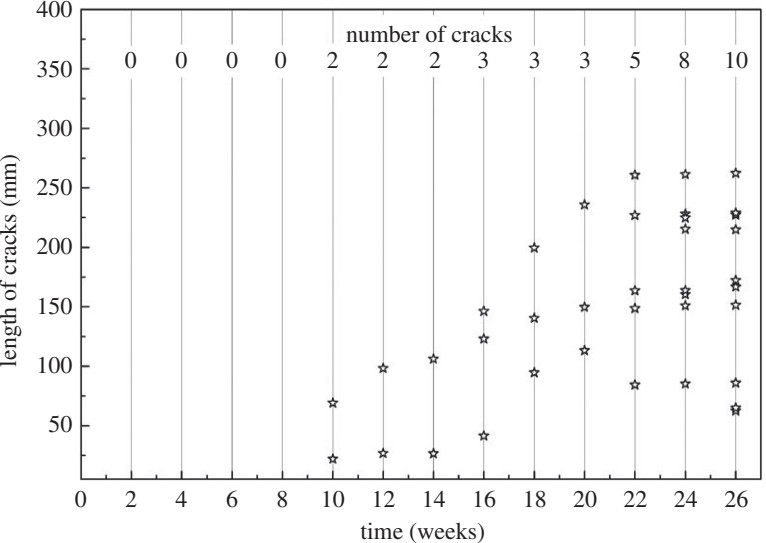

**Figure 7.** Development of cracks on the paraffin-treated bamboo during 26-week outdoor exposure.

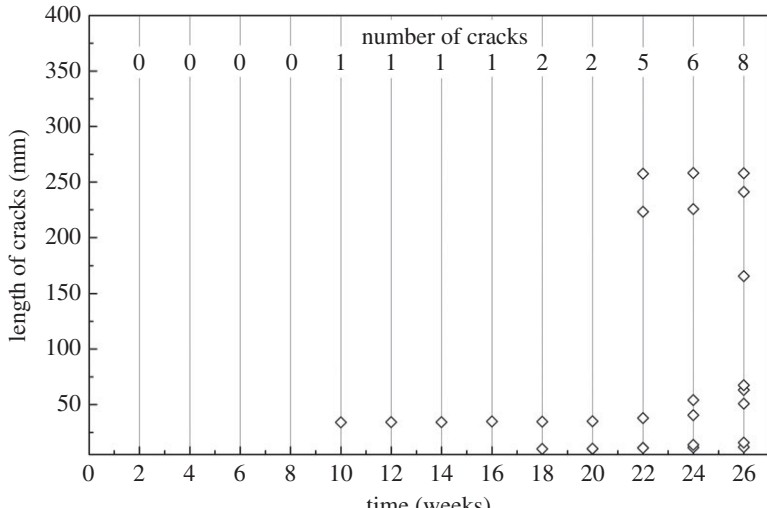

**Figure 8.** Development of cracks on the PEG–PH-treated bamboo during 26-week outdoor exposure.

when exposed outdoors developed cracks after 10 weeks, as seen in figure 7. Comparing with the controls and PEG-treated specimens, the presence of cracks was greatly postponed. In addition, the number of cracks reduced obviously comparing with the controls and PEG treatment. This could be attributed to the formation of the paraffin layer on the surface of round bamboo, which protects bamboo from water and moisture. However, toward the end of the experiment, the length of cracks developed quickly, with four cracks longer than 200 mm, and the reason for which might be that part of the paraffin layer melted, aged and leached due to the variation of environmental conditions (figure 3), leading to the loss of surface protection.

### 3.3.4. Cracks on round bamboo treated with the combination of PEG and paraffin

PEG could improve the dimensional stability of round bamboo by means of bulking the cell wall but was easy to leach out, which became a big problem for outdoor application [22]. As could be seen, the number of cracks on PEG-treated round bamboo increased at the end of the experiment. To form a barrier for PEG-treated bamboo as well as dry bamboo to certain moisture content, paraffin heat treatments were applied. Results in figure 8 revealed that cracking was initially generated at the 10th week; however, only two tiny cracks could be observed before 22 weeks during outdoor exposure. Two new cracks with a length of 200 mm had developed at the 22nd week, which were probably caused by the loss of

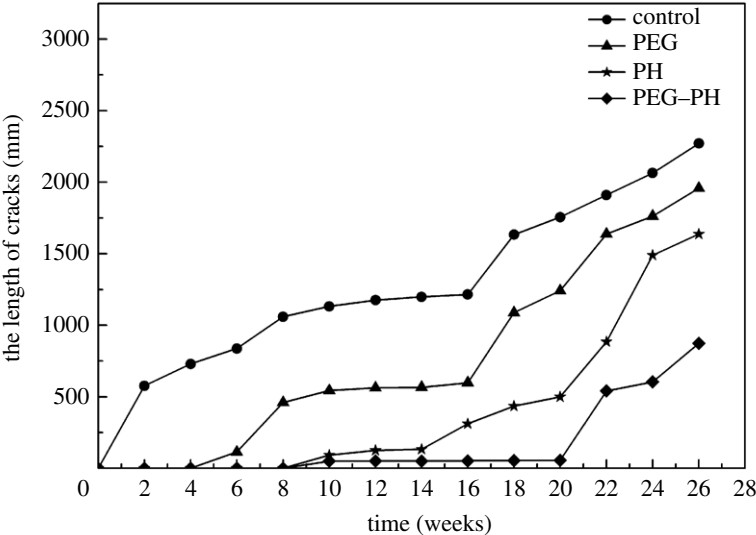

**Figure 9.** The time-dependent change of the crack length with different treatments.

PEG and paraffin protection under a long period of environmental exposure. Eight cracks with three longer than 150 mm developed at the end of the experiment. In general, the combination of PEG immersion and paraffin heat treatment was an effective method for restraining cracks on round bamboo.

### 3.3.5. The time-dependent change of the crack length of each group

To further disclose the effect of the three methods on restraining cracks in round bamboo, the total length of cracks in round bamboo was calculated and results are shown in figure 9. The cracks began to develop after 2nd, 6th, 10th and 22nd weeks, respectively, for the control, PEG, paraffin and PEG–PH treatment. The total lengths of cracks also followed the tendency above, and at the end of the experiment, they were 2271.3, 1957.8, 1637.0, 873.5 mm for the control, PEG, paraffin and PEG–PH treatment. Both PEG and paraffin treatment alone could restrain bamboo from cracking, but far less than PEG–PH treatment. PEG treatment had some effect on preventing cracks but inferior to the method of paraffin treatment. Paraffin heat treatment remarkably decreased the total length of cracking in the earlier period of the experiment but was not so good at the later stage. The combination of the above two methods effectively enhanced the effect of anti-cracking. Obvious cracks that appeared on PEG–PH treatment specimens were greatly postponed, and the total length of cracks was significantly reduced. Based on the above analyses, a conclusion can be made that the combination of PEG impregnation and paraffin heat treatment could effectively prevent round bamboo from cracking.

## 3.4. Hydrophobicity

Bamboo has a strong affinity for water, which caused moisture content to vary with the external temperature and humidity. The change of moisture content is one of the main factors causing cracking. The WCA is a tool used to determine surface properties such as the hydrophobic property. Therefore, the WCA of control and PEG–PH groups was tested and plotted against time.

The contact angle shown in figure 10 displayed that the contact angle of samples treated with PEG alone was 58.7°, much lower than the control owing to the hydrophilicity of PEG, and decreased to 21.2° after 18 s balancing. Paraffin as a hydrophobic chemical gave the sample a higher initial contact angle as high as 113.3°, and the contact angle dropped to 65.0° after 18 s balancing, which was still higher than the control. The initial contact angle of control and PEG–PH groups was 113.0° and 118.5°, respectively. There was no great difference in the initial contact angle of these two. However, there was a significant difference when the contact angle got a balance. The balance contact angle of the control group dropped to 44.5°. By contrast, the balance contact angle of PEG–PH group was 78.9°, much higher than the control group, caused probably by paraffin.

From the analysis, we concluded that the better hydrophobicity of round bamboo could be achieved after PEG–PH treatment, it also indicated that PEG treatment followed by heat treatment with paraffin could effectively reduce the deformation and cracking that were caused by water absorption capacity.

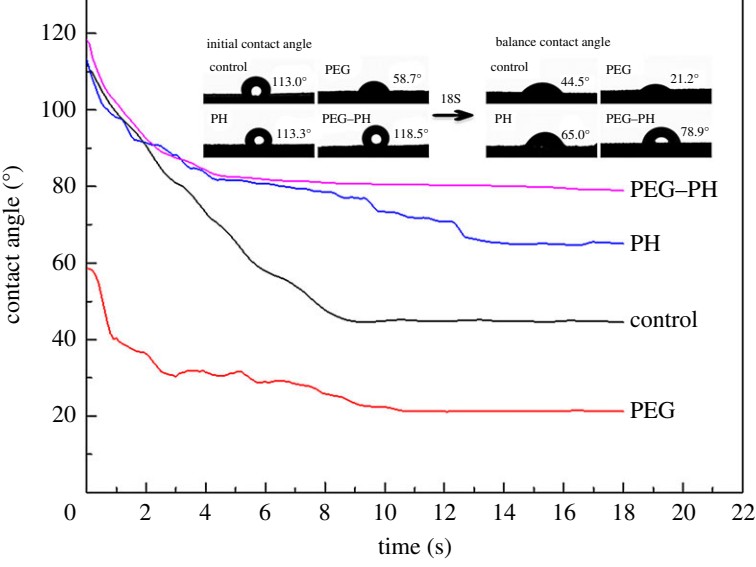

**Figure 10.** The dynamic contact angle of round bamboo with treatment.

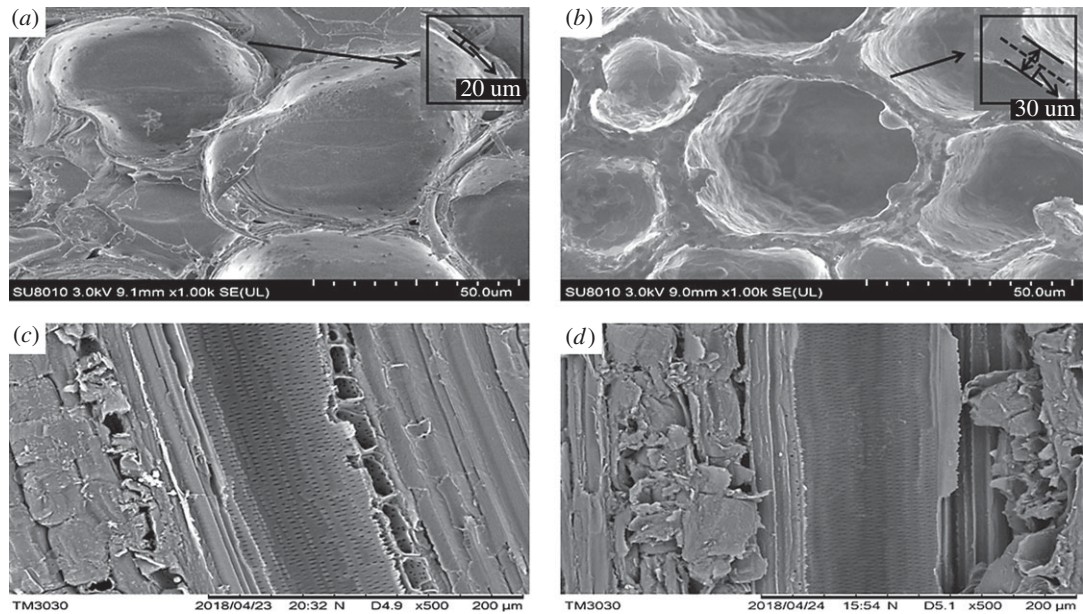

**Figure 11.** SEM images of untreated (*a,c*) and PEG–PH-treated (*b,d*) round bamboo.

### 3.5. SEM analysis

The morphology of bamboo treated with the combination of PEG and paraffin (PEG–PH) was observed by SEM and compared with the untreated bamboo (figure 11). As seen from the transverse section (figure 11*a,b*), the untreated bamboo displayed smooth cell wall, obvious pits and intercellular space. After treatment with PEG–PH, the cell wall became rough, with an intercellular cavity filled and pits covered. Comparing the SEM of untreated bamboo with the PEG–PH-treated one, the cell wall of the latter swelled from 20 to 30 µm and the intercellular cavity disappeared. The radial section also demonstrated that the pits were blocked after PEG–PH treatment as compared with the untreated bamboo (figure 11*c,d*).

The morphology of specimens after 26 weeks of outdoor exposure was also observed by SEM, as shown in figure 12. Large number of mycelia invaded untreated bamboo by accident, penetrating from vessels to the adjacent parenchyma cells, causing obvious pores in the cell wall. Therefore, we suspected that mycelia infestation is one of the reasons which results in cracking, and further exploration will be conducted on the relationship of mycelia infestation and cracking.

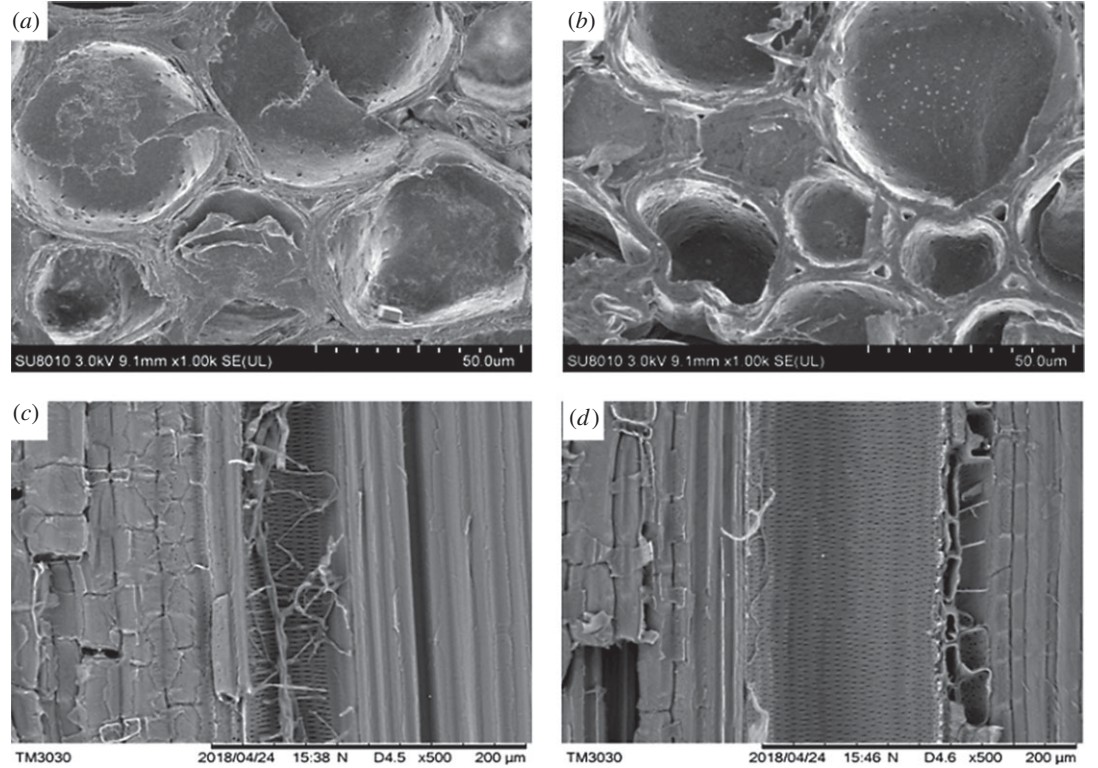

**Figure 12.** SEM images of untreated (*a,c*) and PEG–PH-treated (*b,d*) round bamboo after 26-week outdoor exposure.

The PEG–PH-treated bamboo was slightly affected by fungi and no obvious damage to the neighbouring cells was found. In addition, intense cracks occurred in the untreated bamboo, displaying an approximately concentric pattern within the secondary walls of cells, showing a tendency of cell lamellae separating from each other (figure 12*a*). As for the treated bamboo, cells were densely packed (figure 12*b*) with regular triangle-shaped void spaces in the cell corners. It is hard to distinguish PEG from paraffin by SEM. Samples for SEM prepared from the inner part of bamboo were considered to be filled with PEG alone because the high molecular weight and hydrophobic paraffin is difficult to penetrate bamboo cells. Further research on the microstructure evolution needed to be carried out according to the literature which related to the microstructure characterization of bamboo [23–25].

## 4. Conclusion

To reduce the risks to round bamboo from cracking, three treatments were studied including PEG impregnation, paraffin heating and the combination of PEG impregnation and paraffin heating.

PEG can penetrate fresh round bamboo very well through concentration difference, but the retention of PEG slightly decreased after paraffin heating.

Round bamboo treated with the combination of PEG impregnation and paraffin heat treatment greatly improved the dimensional stability by postponing the initial splitting and reducing the number of cracks and checks. Scanning electron microscopy revealed that the cell wall of bamboo was covered with the treating chemicals, partially blocking the pits, which maintained the cell wall in the original state after 26-week outdoor exposure, as well as restrained the invasion of fungi.

Ethics. This is original work and is part of MA.Sc programme carried out at Zhejiang A&F University, China. All biological samples have been obtained from the bamboo nursery of Anji Zhujing Bamboo Technology Co., Ltd with prior permission. The SEM imaging has been carried out at Zhejiang University, China. Other studies were carried out at Zhejiang A&F University. Besides, I have received informed consent for the participants to participate in this study.
Data accessibility. Our data are deposited at Dryad Digital Repository: https://dx.doi.org/10.5061/dryad.148f85j [26].
Authors' contributions. F.S., W.C. and J.R. designed the research, J.R., J.J. and H.W. collected and analysed the data, interpreted the results and wrote the manuscript. N.K.P., X.Y. and T.L. prepared all samples for analysis. All authors gave final approval for publication.
Competing interests. The authors declare no competing interests.

Funding. Financial support came from the National Natural Science Foundation of China (grant no. 31470587) and the Natural Science Foundation of Zhejiang Province (grant no. LZ14C160002).

Acknowledgements. We deeply appreciate A.P. Zhongqing Ma and his students, School of Engineering at Zhejiang A&F University (China), who interpret how to measure the water contact angle of bamboo. We are also grateful to two anonymous reviewers who provided comments that substantially improved the manuscript.

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
