## [Reviewer comments · Royal Society Open Science]

Review History

RSOS-190105.R0 (Original submission)

Review form: Reviewer 1

Is the manuscript scientifically sound in its present form?

Yes

Are the interpretations and conclusions justified by the results?

Yes

Is the language acceptable?

Yes

Is it clear how to access all supporting data?

Yes

Do you have any ethical concerns with this paper?

No

Have you any concerns about statistical analyses in this paper?

I do not feel qualified to assess the statistics

Recommendation?

Major revision is needed (please make suggestions in comments)

Comments to the Author(s)

The inclination of cracking of round bamboo is a big issue to its application. This study tried to use a combination treatment of PEG and PH to solve this problem. The study is of practical value, while the manuscript needs to be improved in the following aspects:

1. The objectives as proposed in the introduction were not well proved. As suggested in introduction, PEG should penetrate the cell wall and then stabilize the bamboo, and paraffin heat treatment can avoid the leaching of PEG.

For PEG penetration, the authors only gave the retention values of PEG, but no evidence of the PEG distribution in either cell wall or cell cavity. And this distribution can not be easily investigated by SEM observation. The authors should measure the change on thickness of bamboo culm after PEG and PEG-PH treatments, which might give some information on whether PEG entered cell wall or not.

For the effect of PH on leaching of PEG, the authors should compare the PEG retentions of PEG and PEG-PH treated samples after 26 weeks' exposure.

2. In "2.1 Materials", the authors should give more information on the sample preparation method, e.g., how many replicates are used? How the samples were selected and grouped to avoid the big deviations? From Fig.2, it seemed that the samples are quite different in cutting position, namely, the distance to the node. And I think this is quite important to the length of cracking.

3. In the line 2 in 2.3, "liquid paraffin" should be "melted paraffin";

4. In 2.5, the authors used acetone in determination of PEG retention. What is the mechanism of this method? It should be briefly introduce so that the readers could understand.

5. In 2.7, the authors didn't mention how the samples were prepared. Is it carried out on the end surface or not?

6. In 3.1, "The loss of PEG was probably caused by the water soluble property and rinsing procedure during the leaching experiment." What is the leaching experiment? I didn't see any leaching experiment in this study. And also the Paragraph 2 in 3.1 seems no relation with the theme of 3.1.

7. In 3.4, the authors wrote "After treatment with PEG-PH, the cell wall became rough, with intercellular cavity filled and pits covered, which suggested that PEG can swell the cell wall.". The logic in this sentence seems unreasonable. How can the rough cell wall and covered pits suggest the swelling cell wall by PEG? It needs to be revised.

8. The last sentence in 3.4, "The application of paraffin heat-treatment could enhance the resistance of PEG treated round bamboo from fungi by forming a hydrophobic surface protection." This sentence should be reconsidered. Because the types of fungi are so various, and PH can not always improve the fungal resistance. It would be better to find some references to support this conclusion.

Review form: Reviewer 2

Is the manuscript scientifically sound in its present form?

Yes

Are the interpretations and conclusions justified by the results?

Yes

Is the language acceptable?

No

Is it clear how to access all supporting data?

Yes

Do you have any ethical concerns with this paper?

No

Have you any concerns about statistical analyses in this paper?

No

Recommendation?

Major revision is needed (please make suggestions in comments)

Comments to the Author(s)

The manuscript is interesting and raises important issues regarding bamboo conservation. However, overall, the manuscript has to be scrutinized by a native English speaker.

Review form: Reviewer 3

Is the manuscript scientifically sound in its present form?

Yes

Are the interpretations and conclusions justified by the results?

Yes

Is the language acceptable?

Yes

Is it clear how to access all supporting data?

Not Applicable

Do you have any ethical concerns with this paper?

No

Have you any concerns about statistical analyses in this paper?

No

Recommendation?

Major revision is needed (please make suggestions in comments)

Comments to the Author(s)

This paper uses the combination of PEG immersion and paraffin heating to solve the splitting problem for round bamboo. Compared with other existing processes, the method proposed in this paper helps to delay the crack generation. The authors also use SEM and WCA to study the mechanism of crack delay and crack number reduction for round bamboo.

There is a list of things that the author should address and improve.

1. In section 2.4, the author should give more details about the experiment process.
2. It is hard to follow the quantitative analysis of the crack generation and development in section 3.2, including Fig. 6-8. The author should find another way to express the experiment results. For example. The average crack number/length for each treatment.
3. The SEM images are not sufficient to support the author's description. Did the author compare the microstructure evolution of bamboo with different treatments (only PEG, only paraffin heating, and the combination of PEG and paraffin heating)? Did the author check about the SEM images along with the crack initiation during the experiment (such as Habibi et al., Crack propagation in bamboo's hierarchical cellular structure. *Sci Rep*, 2014. 4: 5598)?
4. There is a list of papers related to the microstructure characterization of bamboo as a reference: Habibi et al., Viscoelastic damping behavior of structural bamboo material and its microstructural origins. *Mechanics of Materials*, 2016. 97: 184-198. Dixon PG and Gibson LJ, The structure and mechanics of Moso bamboo material. *Journal of the Royal Society Interface*, 2014. 11: 20140321.

Decision letter (RSOS-190105.R0)

23-May-2019

Dear Miss Rao,

The editors assigned to your paper ("Combination of polyethylene glycol impregnation and paraffin heat treatment to protect round bamboo from cracking") have now received comments from reviewers. We would like you to revise your paper in accordance with the referee and Associate Editor suggestions which can be found below (not including confidential reports to the Editor). Please note this decision does not guarantee eventual acceptance.

Please submit a copy of your revised paper before 15-Jun-2019. Please note that the revision deadline will expire at 00.00am on this date. If we do not hear from you within this time then it will be assumed that the paper has been withdrawn. In exceptional circumstances, extensions may be possible if agreed with the Editorial Office in advance. We do not allow multiple rounds of revision so we urge you to make every effort to fully address all of the comments at this stage. If deemed necessary by the Editors, your manuscript will be sent back to one or more of the original reviewers for assessment. If the original reviewers are not available, we may invite new reviewers.

To revise your manuscript, log into <http://mc.manuscriptcentral.com/rsos> and enter your Author Centre, where you will find your manuscript title listed under "Manuscripts with

Decisions." Under "Actions," click on "Create a Revision." Your manuscript number has been appended to denote a revision. Revise your manuscript and upload a new version through your Author Centre.

- Data accessibility

If you wish to submit your supporting data or code to Dryad (<http://datadryad.org/>), or modify your current submission to dryad, please use the following link:
<http://datadryad.org/submit?journalID=RSOS&manu=RSOS-190105>

- Competing interests

- Authors' contributions

AB carried out the molecular lab work, participated in data analysis, carried out sequence alignments, participated in the design of the study and drafted the manuscript; CD carried out the statistical analyses; EF collected field data; GH conceived of the study, designed the study,

coordinated the study and helped draft the manuscript. All authors gave final approval for publication.

- Acknowledgements

- Funding statement

Kind regards,

Andrew Dunn

on behalf of Prof R. Kerry Rowe (Subject Editor)

Associate Editor's comments:

Three reviewers have been sought for this paper. The editors have made their decision on the basis of the comments received, and recommend a major revision, as it is clear that the manuscript (while promising) has a number of areas that need improvement before it may be considered further. Please ensure that your revision not only identifies (in a comprehensive point-by-point response) all the changes made in response to the reviewer comments but also seek assistance from a language editing service (<https://royalsociety.org/journals/authors/language-polishing/>) to improve the standard of the written English - we sympathise that English is a tricky language, but the onus is on the authors to ensure that the quality of their scientific work is not obscured by the written language.

Comments to Author:

Reviewers' Comments to Author:

Reviewer: 1

Comments to the Author(s)

The inclination of cracking of round bamboo is a big issue to its application. This study tried to use a combination treatment of PEG and PH to solve this problem. The study is of practical value, while the manuscript needs to be improved in the following aspects:

1. The objectives as proposed in the introduction were not well proved. As suggested in introduction, PEG should penetrate the cell wall and then stabilize the bamboo, and paraffin heat treatment can avoid the leaching of PEG.

For PEG penetration, the authors only gave the retention values of PEG, but no evidence of the PEG distribution in either cell wall or cell cavity. And this distribution can not be easily investigated by SEM observation. The authors should measure the change on thickness of bamboo culm after PEG and PEG-PH treatments, which might give some information on whether PEG entered cell wall or not.

For the effect of PH on leaching of PEG, the authors should compare the PEG retentions of PEG and PEG-PH treated samples after 26 weeks' exposure.

2. In "2.1 Materials", the authors should give more information on the sample preparation method, e.g., how many replicates are used? How the samples were selected and grouped to avoid the big deviations? From Fig.2, it seemed that the samples are quite different in cutting position, namely, the distance to the node. And I think this is quite important to the length of cracking.
3. In the line 2 in 2.3, "liquid paraffin" should be "melted paraffin";
4. In 2.5, the authors used acetone in determination of PEG retention. What is the mechanism of this method? It should be briefly introduce so that the readers could understand.
5. In 2.7, the authors didn't mention how the samples were prepared. Is it carried out on the end surface or not?
6. In 3.1, "The loss of PEG was probably caused by the water soluble property and rinsing procedure during the leaching experiment." What is the leaching experiment? I didn't see any leaching experiment in this study. And also the Paragraph 2 in 3.1 seems no relation with the theme of 3.1.
7. In 3.4, the authors wrote "After treatment with PEG-PH, the cell wall became rough, with intercellular cavity filled and pits covered, which suggested that PEG can swell the cell wall. ". The logic in this sentence seems unreasonable. How can the rough cell wall and covered pits suggest the swelling cell wall by PEG? It needs to be revised.
8. The last sentence in 3.4, "The application of paraffin heat-treatment could enhance the resistance of PEG treated round bamboo from fungi by forming a hydrophobic surface protection." This sentence should be reconsidered. Because the types of fungi are so various, and PH can not always improve the fungal resistance. It would be better to find some references to support this conclusion.

Reviewer: 2

Comments to the Author(s)

The manuscript is interesting and raises important issues regarding bamboo conservation. However, overall, the manuscript has to be scrutinized by a native English speaker.

Reviewer: 3

Comments to the Author(s)

This paper uses the combination of PEG immersion and paraffin heating to solve the splitting problem for round bamboo. Compared with other existing processes, the method proposed in this paper helps to delay the crack generation. The authors also use SEM and WCA to study the mechanism of crack delay and crack number reduction for round bamboo.

There is a list of things that the author should address and improve.

1. In section 2.4, the author should give more details about the experiment process.
2. It is hard to follow the quantitative analysis of the crack generation and development in section 3.2, including Fig. 6-8. The author should find another way to express the experiment results. For example. The average crack number/length for each treatment.
3. The SEM images are not sufficient to support the author's description. Did the author compare

the microstructure evolution of bamboo with different treatments (only PEG, only paraffin heating, and the combination of PEG and paraffin heating)? Did the author check about the SEM images along with the crack initiation during the experiment (such as Habibi et al., Crack propagation in bamboo's hierarchical cellular structure. *Sci Rep*, 2014. 4: 5598)?

4. There is a list of papers related to the microstructure characterization of bamboo as a reference: Habibi et al., Viscoelastic damping behavior of structural bamboo material and its microstructural origins. *Mechanics of Materials*, 2016. 97: 184-198. Dixon PG and Gibson LJ, The structure and mechanics of Moso bamboo material. *Journal of the Royal Society Interface*, 2014. 11: 20140321.

Author's Response to Decision Letter for (RSOS-190105.R0)

See Appendix A.

RSOS-190105.R1 (Revision)

Review form: Reviewer 2

Is the manuscript scientifically sound in its present form?

Yes

Are the interpretations and conclusions justified by the results?

Yes

Is the language acceptable?

Yes

Do you have any ethical concerns with this paper?

No

Have you any concerns about statistical analyses in this paper?

No

Recommendation?

Accept as is

Comments to the Author(s)

It seems that the Authors corrected the most important drawbacks of the paper making it more clear and easier to read.

Review form: Reviewer 3

Is the manuscript scientifically sound in its present form?

Yes

Are the interpretations and conclusions justified by the results?

Yes

Is the language acceptable?

Yes

Do you have any ethical concerns with this paper?

No

Have you any concerns about statistical analyses in this paper?

No

Recommendation?

Accept as is

Comments to the Author(s)

The authors have addressed my previous comments very well. The manuscript can be published as is.

Review form: Reviewer 4

Is the manuscript scientifically sound in its present form?

Yes

Are the interpretations and conclusions justified by the results?

Yes

Is the language acceptable?

Yes

Do you have any ethical concerns with this paper?

No

Have you any concerns about statistical analyses in this paper?

No

Recommendation?

Accept with minor revision (please list in comments)

Comments to the Author(s)

2.1. four samples with one node, and the other two with two nodes, then is there any differences in cracking resistance between them?

3.4 Figure 10.

will you please add the dynamic contact angle of bamboo treated with paraffin? just as figure 9. did. It will be much better to keep the comparisons consistent, and helpful to explain the cracking behaviors.

Decision letter (RSOS-190105.R1)

24-Sep-2019

Dear Miss Rao:

On behalf of the Editors, I am pleased to inform you that your Manuscript RSOS-190105.R1 entitled "Combination of polyethylene glycol impregnation and paraffin heat treatment to protect round bamboo from cracking" has been accepted for publication in Royal Society Open Science subject to minor revision in accordance with the referee suggestions. Please find the referees' comments at the end of this email.

The reviewers and Subject Editor have recommended publication, but also suggest some minor revisions to your manuscript. Therefore, I invite you to respond to the comments and revise your manuscript.

- Ethics statement

- Data accessibility

If you wish to submit your supporting data or code to Dryad (<http://datadryad.org/>), or modify your current submission to dryad, please use the following link:
<http://datadryad.org/submit?journalID=RSOS&manu=RSOS-190105.R1>

- Competing interests

- Authors' contributions

All submissions, other than those with a single author, must include an Authors' Contributions section which individually lists the specific contribution of each author. The list of Authors should meet all of the following criteria; 1) substantial contributions to conception and design, or

acquisition of data, or analysis and interpretation of data; 2) drafting the article or revising it critically for important intellectual content; and 3) final approval of the version to be published.

- Acknowledgements

- Funding statement

Because the schedule for publication is very tight, it is a condition of publication that you submit the revised version of your manuscript before 03-Oct-2019. Please note that the revision deadline will expire at 00.00am on this date. If you do not think you will be able to meet this date please let me know immediately.

- 1) A text file of the manuscript (tex, txt, rtf, docx or doc), references, tables (including captions) and figure captions. Do not upload a PDF as your "Main Document".
- 2) A separate electronic file of each figure (EPS or print-quality PDF preferred (either format should be produced directly from original creation package), or original software format)
- 3) Included a 100 word media summary of your paper when requested at submission. Please ensure you have entered correct contact details (email, institution and telephone) in your user account

4) Included the raw data to support the claims made in your paper. You can either include your data as electronic supplementary material or upload to a repository and include the relevant doi within your manuscript

5) All supplementary materials accompanying an accepted article will be treated as in their final form. Note that the Royal Society will neither edit nor typeset supplementary material and it will be hosted as provided. Please ensure that the supplementary material includes the paper details where possible (authors, article title, journal name).

on behalf of Prof R. Kerry Rowe (Subject Editor)
openscience@royalsociety.org

Associate Editor Comments to Author:

A couple of minor queries remain from one of the reviewers, but otherwise this manuscript appears to be ready for acceptance. Please respond to the queries, and make changes required by the reviewers before resubmitting.

Associate Editor: 2
Comments to the Author:
(There are no comments.)

Reviewer comments to Author:
Reviewer: 3

Comments to the Author(s)
The authors have addressed my previous comments very well. The manuscript can be published as is.

Reviewer: 4

Comments to the Author(s)
2.1. four samples with one node, and the other two with two nodes,

then is there any differences in cracking resistance between them?

3.4 Figure 10.

will you please add the dynamic contact angle of bamboo treated with paraffin? just as figure 9. did. It will be much better to keep the comparisons consistent, and helpful to explain the cracking behaviors.

Reviewer: 2

Comments to the Author(s)

It seems that the Authors corrected the most important drawbacks of the paper making it more clear and easier to read.

Author's Response to Decision Letter for (RSOS-190105.R1)

See Appendix B.

Decision letter (RSOS-190105.R2)

02-Oct-2019

Dear Miss Rao,

I am pleased to inform you that your manuscript entitled "Combination of polyethylene glycol impregnation and paraffin heat treatment to protect round bamboo from cracking" is now accepted for publication in Royal Society Open Science.

on behalf of Prof R. Kerry Rowe (Subject Editor)
openscience@royalsociety.org

Follow Royal Society Publishing on Twitter: [@RSocPublishing](https://twitter.com/RSocPublishing)
Follow Royal Society Publishing on Facebook:
<https://www.facebook.com/RoyalSocietyPublishing.FanPage/>
Read Royal Society Publishing's blog: <https://blogs.royalsociety.org/publishing/>

Appendix A

Thank you very much for your email which you sent us the reviewer's report on our paper, we also wish to take this opportunity to thank the reviewer for constructive comments and valuable recommendations. We have studied the comments carefully and tried our best to revise the manuscript according to those good comments. Revised words are marked in red in the paper. The main corrections in the paper and the responds to the reviewer's comments are as follows:

Response to Reviewer 1:

1. The objectives as proposed in the introduction were not well proved. As suggested in introduction, PEG should penetrate the cell wall and then stabilize the bamboo, and paraffin heat treatment can avoid the leaching of PEG. For PEG penetration, the authors only gave the retention values of PEG, but no evidence of the PEG distribution in either cell wall or cell cavity. And this distribution can not be easily investigated by SEM observation. The authors should measure the change on thickness of bamboo culm after PEG and PEG-PH treatments, which might give some information on whether PEG entered cell wall or not.

Response: Thanks for your good suggestion. The material was fresh round bamboo that was in water saturated situation in this study. PEG permeated into bamboo by displacing the liquid moisture. It wasn't applicable to analyze the PEG distribution according to the change on thickness of bamboo culm. Therefore, we determined the amount of PEG in bamboo by extraction referring to the method from Kang et al. We also measure the change on thickness of bamboo culm after PEG and PEG-PH treatments by SEM, as figure11 shown. Comparing the SEM of untreated bamboo(a) with the PEG-PH treated one(b), the cell wall of the later swelled from 20 μm to 30 μm and the intercellular cavity disappeared, which indicated that PEG penetrated into the cell wall or cell cavity.

Figure 11. SEM images of untreated (a and c) and PEG-PH treated (b and d) round bamboo.

2. For the effect of PH on leaching of PEG, the authors should compare the PEG retentions of PEG and PEG-PH treated samples after 26 weeks' exposure.

Response: Thanks for your suggestion, we will study the retention and leaching of PEG in our future experiments. In this paper, we focus on the effect of PEG and PEG-PH treatment in preventing round bamboo from cracking both in the treatment procedure and

moistening/water-drying procedure, especially the latter.

The application of paraffin heat treatment after PEG immersion is to explore a suitable method for drying round bamboo. Round bamboo is lack in drying technology, nor having an advanced drying schedule like wood. The commonly adopt air drying procedure is not only time-consuming, but also cause lots of cracks. We have selected the best heating procedure from the preliminary experiments and applied in this paper.

The main aim of paraffin heat treatment was to dry round bamboo. The 3.2 section in resubmitted manuscript is to analysis if the paraffin heat treatment procedure caused the movement and loss of PEG during water evaporation. The results shown that PEG slightly lost after paraffin heat treatment. Additionally, the formation of paraffin coating on the surface of round bamboo in PEG-PH treatment provide hydrophobicity to bamboo, which prevent bamboo from water, as well as PEG from leaching. In our future researches, we will use cross-linking or chemical modification to prevent PEG from leaching out of round bamboo. We revised 3.2 section in revised manuscript, please check.

3. In “2.1 Materials”, the authors should give more information on the sample preparation method, e.g., how many replicates are used? How the samples were selected and grouped to avoid the big deviations?

From Fig.2, it seemed that the samples are quite different in cutting position, namely, the distance to the node. And I think this is quite important to the length of cracking.

Response: Accepted. Indeed, the diameter and the number of the node would affect the length of cracking, we hadn't analyzed so comprehensive as reviewer said. We have added sample details in section 2.1. Please check. The sample details added were as follows:

Freshly cut five-year old *Phyllostachys iridescens* was obtained from Anji, Zhejiang, China. The round bamboo with diameter of 42-61 mm was cut into specimens with a length of 300 mm. The average moisture content of the specimens was $58.3\pm 5\%$, and six replicates were prepared for each treatment. To avoid large deviation in every group, the diameter of four samples were in the range of 42-50 mm with one node, and the other two were in 50-61 mm with two nodes at each end.

4. In the line 2 in 2.3, “liquid paraffin” should be “melted paraffin”.

Response: Accepted. We have changed all “liquid paraffin” to “melted paraffin” in resubmitted manuscript, please check.

5. In 2.5, the authors used acetone in determination of PEG retention. What is the mechanism of this method? It should be briefly introduced so that the readers could understand.

Response: Accepted. We added the mechanism of this method in section 2.5. Please check. The details added were as follows:

Owing to the high moisture content of fresh round bamboo, the bulking coefficient was difficult to determine according to the commonly adopted formula [7]. Acetone has lower boiling point, dissolves PEG easily and is insoluble in paraffin, hence, it can be used to extract PEG from bamboo.

6. In 2.7, the authors didn't mention how the samples were prepared. Is it carried out on the end surface or not?

Response: Accepted. Determination of Surface Contact Angle was conducted on the end section of round bamboo, we have complemented this detail of sampling in 2.7. please check.

7. In 3.1, “The loss of PEG was probably caused by the water soluble property and rinsing procedure during the leaching experiment.” What is the leaching experiment? I didn’t see any leaching experiment in this study.

Response: Accepted. Thanks for your good suggestion. The sentence “The loss of PEG was probably caused by the water soluble property and rinsing procedure during the leaching experiment.” is unsuitable in 3.1, we deleted this sentence. In our study, field test was carried out to test the cracking of round bamboo. As there is no standard method for testing bamboo modifying agents, we take the standards of wood preservatives as reference. According to the AWPA E18-13 “Standard field test for evaluation of wood preservatives to be used above ground (uc3b); ground proximity decay test” , there’s no requirement for leaching experiments. In this paper, we didn’t do the leaching experiment. We only studied the content of PEG in round bamboo after PEG and PEG-PH treatment. We had revised this section so that the readers could understand, the analysis results were as follows:

3.2 The content of PEG

Paraffin heat treatment can form hydrophobic surface for PEG treated round bamboo, but probably cause the loss of PEG due to the movement of PEG in water evaporation. Therefore, the amount of PEG will be determined. The average content of PEG was 17.0 (± 0.8) g/kg for PEG treatment alone, while 15.3(± 1.7) g/kg for the combination of PEG and paraffin treatment. It seemed that PEG slightly lost after the paraffin heat treatment, which was probably caused by the movement of PEG with the evaporation of water.

And also the Paragraph 2 in 3.1 seems no relation with the theme of 3.1.

Response: So sorry for this, we have set paragraph 2 as one of the sections in revised manuscript title as “3.1 Type of crack in round bamboo”, please check.

8. In 3.4, the authors wrote “After treatment with PEG-PH, the cell wall became rough, with intercellular cavity filled and pits covered, which suggested that PEG can swell the cell wall. “. The logic in this sentence seems unreasonable. How can the rough cell wall and covered pits suggest the swelling cell wall by PEG? It needs to be revised.

Response: We measured the thickness of cell wall before and after impregnating PEG, and the data of thickness were added in Fig.11. Please check Fig.11 in section 3.5. Besides, the revised analysis results from SEM were placed in resubmitted manuscript and notes. The new analysis results are as follows:

After treatment with PEG-PH, the cell wall became rough, with intercellular cavity filled and pits covered. Comparing the SEM of untreated bamboo with the PEG-PH treated one, the cell wall of the later swelled from 20 μm to 30 μm and the intercellular cavity disappeared.

Figure 11. SEM images of untreated (a and c) and PEG-PH treated (b and d) round bamboo.

9. The last sentence in 3.4, “The application of paraffin heat-treatment could enhance the resistance of PEG treated round bamboo from fungi by forming a hydrophobic surface protection.” This sentence should be reconsidered. Because the types of fungi are so various, and PH can not always improve the fungal resistance. It would be better to find some references to support this conclusion.

Response: Accepted. We are very sorry for this. PH treatment can not always improve the fungal resistance as reviewer said above, so we deleted this sentence. The new analysis results were as follows:

Further researches on the microstructure evolution needed to be carried out according to the literatures which related to the microstructure characterization of bamboo[23-25].”Please check the last sentence in section 3.5 in resubmitted manuscript.

Response to Reviewer 2:

The manuscript is interesting and raises important issues regarding bamboo conservation. However, overall, the manuscript has to be scrutinized by a native English speaker.

Response: Special thanks to you for your good comments, we have asked a native English speaker to improve the standard of the written English.

Response to Reviewer 3:

1. In section 2.4, the author should give more details about the experiment process.

Response: Thanks for your good suggestion. in section 2.4, it is a combination of two treating procedure. The first procedure is PEG immersion and the second is paraffin heat treatment. The detailed description of the experiment process was added in section 2.2 and 2.3 respectively. Please check.

2. It is hard to follow the quantitative analysis of the crack generation and development in section 3.2, including Fig. 6-8. The author should find another way to express the experiment results. For example. The average crack number/length for each treatment.

Response: Thanks for your good suggestion. In fact, we put forward some ways to evaluate the cracks or splits of round bamboo, such as length, width, depth, or number of cracks. In actual measurements, only the length and number could be used to evaluate the cracks accurately, so we demonstrated the number and length of cracks developed during 26-week exposure. When we used the average length of crack (Eq.1) to evaluate the overall effect of each treatment, we found it could not reflect the real effect of each treatment, because with the same total length, the more the number of cracks, the lower the average length.

$$\text{the average length of cracks} = \frac{\text{the total length of cracks}}{\text{the number of cracks}} \quad \text{Eq.1}$$

3. The SEM images are not sufficient to support the author's description. Did the author compare the microstructure evolution of bamboo with different treatments (only PEG, only paraffin heating, and the combination of PEG and paraffin heating)? Did the author check about the SEM images along with the crack initiation during the experiment (such as Habibi et al., Crack propagation in bamboo's hierarchical cellular structure. *Sci Rep*, 2014. 4: 5598).

Response: Thanks for your suggestion and good literature, and we have included it in our paper. The literatures are good helps in analyzing the SEM images.

We have not done such detailed research in cellular structure in this paper. The main purpose of this study was to investigate the effect of PEG-PH treatment, so we didn't compare the microstructure evolution of bamboo with different treatments. Besides, we mainly researched the development of macroscopic cracks in this study, and hadn't studied so detailed yet as reviewer said. We will compare the formation and development of cracks in the untreated and treated round bamboo in detail from the micro level in our future research referring to methods reported by Habibi et al.

Special thanks to you for the papers related to the microstructure characterization of bamboo. The papers were a great help to our further researches in the microstructure evolution of round bamboo.

4. There is a list of papers related to the microstructure characterization of bamboo as a reference: Habibi et al., Viscoelastic damping behavior of structural bamboo material and its microstructural origins. *Mechanics of Materials*, 2016. 97: 184-198. Dixon PG and Gibson LJ, The structure and mechanics of Moso bamboo material. *Journal of the Royal Society Interface*, 2014. 11: 20140321.

Response: Thank you for the references, these references are a great help to analyze the microstructure of bamboo, as well as to our following research. We cited these references in resubmitted manuscript, please check the last paragraph on page 11.

Appendix B

Dear editors of Royal Society Open Science:

Thanks so much for your notes and suggestions, we also wish to take this opportunity to thank the reviewers for their positive and constructive comments.

We have revised the manuscript, according to the comments and suggestions of reviewers and editor. The main corrections in the paper and the responds to the reviewer's comments are as follows:

Response to Reviewer 4:

1.2.1. four samples with one node, and the other two with two nodes, then is there any differences in cracking resistance between them?

Response: Thanks for your good suggestion. We will focus on the relationship between the amount and position of nodes and cracking in our future studies, including both the pure round bamboo and the treated ones.

In this paper, we didn't do special statistics on the differences in cracking resistance between the one node and two nodes. This research is interested in the cracking situation of chemically treated and untreated round bamboo.

2. 3.4 Figure 10. will you please add the dynamic contact angle of bamboo treated with paraffin? just as figure 9. did. It will be much better to keep the comparisons consistent, and helpful to explain the cracking behaviors.

Response: Accepted. We have added the dynamic contact angle of bamboo treated with paraffin(The figure 10 has been revised). The 3.4 part was also revised according to the updated figure, please check.

Figure 10. The dynamic contact angle of round bamboo with treatment

Kind regards,
Jin Rao